Growth and metal bioconcentration by conspecific freshwater macroalgae cultured in industrial waste water

Ellison Michael B.
de Nys Rocky
Paul Nicholas A.
Roberts David A. david.roberts1@jcu.edu.au
MACRO—the Centre for Macroalgal Resources and Biotechnology, and School of Marine and Tropical Biology, James Cook University , Townsville, Queensland , Australia
Johnson Stephen
Electronic publication date: 2014 May 22
Publication date: 2014
Volume: 2
Electronic Location ID: e401
Received 2014 Mar 10; Accepted 2014 May 7
Copyright: © 2014 Ellison et al.
Copyright year: 2014
Copyright holder: Ellison et al.
License: This is an open access article distributed under the terms of the Creative Commons Attribution License, which permits unrestricted use, distribution, reproduction and adaptation in any medium and for any purpose provided that it is properly attributed. For attribution, the original author(s), title, publication source (PeerJ) and either DOI or URL of the article must be cited.
License URL: https://creativecommons.org/licenses/by/4.0/

Keywords: Bioremediation, Algae, Waste water, Metals, Metalloids, Coal

Funding: MBD Energy Research and Development program for Biological Carbon Capture and Storage Advanced Manufacturing Cooperative Research Centre (AMCRC) Australian Renewable Energy Agency (ARENA) This research is part of the MBD Energy Research and Development program for Biological Carbon Capture and Storage with the cooperation of Stanwell Energy Corporation. This project is supported by the Advanced Manufacturing Cooperative Research Centre (AMCRC), funded through the Australian Government’s Cooperative Research Centre Scheme, and the Australian Renewable Energy Agency (ARENA). The funders had no role in study design, data collection and analysis, decision to publish, or preparation of the manuscript.

==============================
The bioremediation of industrial waste water by macroalgae is a sustainable and renewable approach to the treatment of waste water produced by multiple industries. However, few studies have tested the bioremediation of complex multi-element waste streams from coal-fired power stations by live algae. This study compares the ability of three species of green freshwater macroalgae from the genus Oedogonium, isolated from different geographic regions, to grow in waste water for the bioremediation of metals. The experiments used Ash Dam water from Tarong power station in Queensland, which is contaminated by multiple metals (Al, Cd, Ni and Zn) and metalloids (As and Se) in excess of Australian water quality guidelines. All species had consistent growth rates in Ash Dam water, despite significant differences in their growth rates in “clean” water. A species isolated from the Ash Dam water itself was not better suited to the bioremediation of that waste water. While there were differences in the temporal pattern of the bioconcentration of metals by the three species, over the course of the experiment, all three species bioconcentrated the same elements preferentially and to a similar extent. All species bioconcentrated metals (Cu, Mn, Ni, Cd and Zn) more rapidly than metalloids (As, Mo and Se). Therefore, bioremediation in situ will be most rapid and complete for metals. Overall, all three species of freshwater macroalgae had the ability to grow in waste water and bioconcentrate elements, with a consistent affinity for the key metals that are regulated by Australian and international water quality guidelines. Together, these characteristics make Oedogonium a clear target for scaled bioremediation programs across a range of geographic regions.

Introduction

The cultivation of algae in industrial effluent provides an effective form of bioremediation as algae can bioconcentrate metals from waste water (Mehta & Gaur, 2005; Troell et al., 2009; Hubbe, Hasan & Ducoste, 2011) while also capturing carbon and producing sustainable biomass (Roberts, de Nys & Paul, 2013). Integrated algal culture is particularly suited to coal-fired power stations as they are significant sources of both metal-contaminated waste water and CO2 (Roberts, de Nys & Paul, 2013). Water and CO2 are two critical constraints to sustainable and cost effective aquatic biomass production (McGinn et al., 2011; Slade & Bauen, 2013). Therefore, integrating algal cultivation with coal-fired power stations provides a path to biomass production with concomitant bioremediation.

The success of the integrated aquaculture model relies on identifying species of algae that are able to satisfy a number of criteria. Target species should preferably be robust to a range of environmental conditions, competitively dominant, and widely distributed. Alternatively, robust local strains are required to be isolated for region-specific applications and avoid the introduction of non-endemic species. Green freshwater macroalgae are particularly strong candidates for scaled bioremediation and biomass production as they satisfy these criteria (Lawton, de Nys & Paul, 2013). In addition, macroalgae require little specialised equipment to harvest and process, and are able to bioconcentrate a wide range of metal ions from solution (Saunders et al., 2012; Roberts, de Nys & Paul, 2013; Lawton, de Nys & Paul, 2013). Previous experimental research has shown that species of the freshwater genus of macroalgae, Oedogonium, can maintain high productivities in “clean” water monocultures, readily outcompete co-occurring species under a variety of environmental conditions, and are widely distributed throughout temperate, sub-tropical and tropical climates in Australia (Lawton et al., 2014). Oedogonium is a genus of unbranched filamentous algae with a worldwide distribution and grows attached to the substrate or as free-floating mats. The widespread distribution of the genus, coupled with a filamentous growth form that is easy to harvest, make species within the genus key candidates for scaled bioremediation and biomass production.

The maintenance of high growth rates in biomass production is the cornerstone to developing effective algal-based bioremediation (Roberts, de Nys & Paul, 2013). There can be substantial variation in growth rates between conspecific species of Oedogonium in small-scale laboratory cultures (Lawton et al., 2014). However, it is unclear whether variation exists between conspecific species in contaminated waste water and whether growth rates in clean water translates to similar growth rates in waste water, and the bioconcentration of metals therein. The bioconcentration of metal ions by macroalgae is a two phase process involving adsorption (rapid and passive sorption of metal ions to cellular surfaces) followed by absorption (metabolically facilitated internalisation of metal ions) (Genter, 1996; Volesky, 2007). The rate of bioremediation of metals from industrial waste waters therefore varies depending upon the growth rate of the species, with high growth rates having the dual benefit of delivering more rapid bioremediation and more biomass for end-use applications (Roberts, de Nys & Paul, 2013). Macroalgae have an ability to tolerate high metal concentrations in water through the production of metal-binding phytochelatins (Pawlik-Skowrońska, 2001) and polyphosphate bodies (Nishikawa et al., 2003), or the sequestration of metals in storage vacuoles (Hanikenne et al., 2005). While these are ubiquitous properties of marine and freshwater species alike, co-occurring conspecifics can have significantly different metal profiles in situ (Brown, Hodgkinson & Hurd, 1999; Sawidis et al., 2001; Roberts, Johnston & Poore, 2008) demonstrating the potential for significant variations in the suitability of conspecific species for bioremediation applications.

We have previously demonstrated that Oedogonium can be cultivated in industrial waste water from a coal-fired power station for the purposes of carbon capture and bioconcentration of a range of metals (Al, Cd, Ni and Zn) and metalloids (As and Se for the purposes of this study) using a locally isolated species (Roberts, de Nys & Paul, 2013). However, a key uncertainty is whether other clearly differentiated species within the genus Oedogonium from different geographic locations have a similar capacity to grow and bioconcentrate metals and metalloids from waste water to provide a bioremediation service. In this study we build upon previous experimental research by measuring the growth rates and bioremediation of metals from a real-world industrial effluent using multiple species of Oedogonium. We address two key questions. First, do species of Oedogonium have a consistent ability to grow in metal-contaminated waste water from a coal fired power station? Second, do species of Oedogonium bioconcentrate a similar quantity and composition of metals from complex waste water? Together, these data provide a basis to begin to assess the suitability of the genus Oedogonium more generally as a target for scaled bioremediation programs across broad geographic regions.

Materials and Methods

Biomass and effluent collection

Three species of filamentous algae from the genus Oedogonium were used in the cultivation experiments. One species was collected from the Ash Dam water (ADW) storage at Tarong power station in October 2012 (‘Tarong Oedogonium’), while the other two species were collected from irrigation ditches in the Brandon sugarcane region. Attempts at species level identification of the three species were made using taxonomic keys (Entwisle et al., 2007), however, the lack of clear morphological characteristics meant that the species could not be identified beyond genus level. The species were therefore assessed using molecular techniques, arguably the most effective approach to identifying cryptic species, and each was determined to be a unique genotype, supporting its assignment as a unique species (Lawton et al., 2014). As the three isolates could not be matched to extant species of Oedogonium on the basis of classical taxonomic features they are hereafter referred to by their GenBank accession numbers. The Tarong Oedogonium isolate (26.76°S, 151.92°E; temperate) has the accession number KC606974, while the two strains isolated from the Brandon region (19.56°S, 147.36°E; tropical) in far north Queensland, have the accession numbers KC701473 and KF606977. The strains from the Brandon region were first collected in 2010 and have been maintained in culture at James Cook University (JCU) for more than 2 years. All three species have been maintained in stock cultures at the Marine and Aquaculture Research Faculties Unit (MARFU), JCU, Douglas campus (19.33°S, 146.76°E), in dechlorinated town water (DTW) with f/2 media addition since collection.

The ADW used in this study is an effluent created during ash disposal practices at a coal-fired power station. ADW was sourced directly from the Tarong coal-fired power station in south-east Queensland (26.76°S, 151.92°E) and transported to James Cook University (JCU), Townsville in clean plastic 1000 L Intermediate Bulk Containers (IBCs) in November 2012. The ADW was then stored at ambient temperature in 12,000 L storage tanks. A sub-sample of this water was taken from the storage tank for this study in December 2012 and subject to elemental analysis to quantify initial conditions in the effluent (Table 1). The ADW was provided by Stanwell Energy Corporation.

Table 1 Elemental composition of dechlorinated town water and Ash Dam water.

Element	Dechlorinated town
water (µg L−1 ± SE)	Ash Dam water
(µg L−1 ± SE)	95A% ANZECC trigger
value (µg L−1)	
Aluminium	10 ± 5.8	123.3 ± 13.3a	55	
Arsenic	1 ± 0.6	33.7 ± 1.2a	24	
Barium	<LOD	<LOD	ID	
Cadmium	0.1 ± 0.1	2.5 ± 0.0a	0.2	
Chromium	<LOD	<LOD	1.0	
Cobalt	<LOD	<LOD	ID	
Copper	2 ± 1.2	1 ± 0.0	1.4	
Iron	50 ± 58.9	50 ± 0.0	ID	
Lead	1 ± 0.6	1 ± 0.0	3.4	
Magnesium	2 ± 1.2	92.7 ± 0.7a	ID	
Manganese	1 ± 0.6	4 ± 0.0a	1900	
Molybdenum	1 ± 0.6	1280 ± 35.1a	ID	
Nickel	1 ± 0.6	34.7 ± 0.3a	11	
Selenium	10 ± 5.8	70 ± 0.0a	11	
Strontium	56.7 ± 32.7	2243 ± 63.3a	ID	
Vanadium	10 ± 5.8	843.3 ± 17.6a	ID	
Zinc	8 ± 4.6	55 ± 0.6a	8.0	
Notes.

a Significantly greater than dechlorinated town water (two-tailed t-test, P < 0.05).

<LOD, less than Level of Detection (1 µg L−1 for most elements). ID, insufficient data to calculate ANZECC 95% trigger values. All data are mean concentrations ± S.E. Bold values exceed the ANZECC trigger value.

Experimental design

A cultivation study was conducted to assess the growth of the three species of Oedogonium, as well as the bioconcentration of metals and metalloids from the ADW. The experiments were conducted within a climate controlled room (28°C, 100 µmol photons m−2 s−1 and a 12 light: 12 dark photoperiod). The experimental temperature was chosen to reflect the median daytime temperature at the time of running the experiments. Stock cultures of each species were maintained for five weeks in DTW with f/2 nutrient addition (0.1 g Aquasonic® f/2 growth media L−1) to acclimate them to the experimental conditions. The aim of the acclimation period was to attain ‘steady state’ productivity, defined here as <5% change in Specific Growth Rate (SGR) for two consecutive weeks prior to initiating the experiment. This was achieved on day 35 (Fig. S1) , and from this time point onwards the three species were cultivated in either Ash Dam water or DTW with f/2 addition as described below, for three consecutive growth periods of seven days. The experimental period therefore spanned 21 days and consisted of three cultivation and harvest cycles of equal duration as described below.

The three species of Oedogonium were cultured in 1 L Schott bottles at an initial stocking density 0.5 g fresh weight (FW) L−1. The FW was measured by gently blotting surface moisture from the biomass and weighing it. The Schott bottles were stocked with biomass from one of the three species and filled with either ADW or DTW amended with f/2 media, and then placed randomly in the climate controlled room. The bottles were aerated with compressed air delivered via a Pasteur pipette (at a flow rate of 0.2 L min−1) to each replicate bottle, and they were randomly redistributed on a daily basis to avoid possible light bias in the experimental area. Each replicate bottle was harvested every seven days by pouring the contents through 20 µm filter mesh. A sample of the water was filtered (0.45 µm) and retained for metal analysis. The biomass was blot-dried with paper towel and weighed to the nearest 0.1 g. A sub-sample of this biomass was returned to the bottle with new water and growth media to reset the stocking density at 0.5 g FW L−1. Any remaining biomass from each harvest was then dried in a dehydrator for 48 h at 60 °C.

Growth rates for each of the treatments were calculated based upon the FW determined for eight consecutive growth periods of seven days (t = Day 7). The Specific Growth Rate, SGR (% FW d−1), was calculated using the equation: SGR=LnWf/Wi/T∗100

where Wf, is the final weight (g FW) of biomass, Wi, is the initial weight (g FW) of biomass, and T, is the number of days in culture (7).

Elemental analysis

The algal biomass and the two water sources (ADW and DTW) were each analysed for the same 17 elements at the start of the experiment (Table 1). Water samples were collected using a 60 mL syringe and filtered (0.45 µm) to remove particulates. The concentrations of the 17 elements (Table 1) were also determined for all three species of Oedogonium grown in the Ash Dam water treatments before exposure to Ash Dam water, and at each of the three time points during the 21 day experimental period when Oedogonium was cultivated in Ash Dam water. All biomass was prepared for the analysis by drying in a dehydrator for 48 h at 60 °C, then digesting 100 milligrams (mg) of the dried algae in a Teflon digestion vessel with 3.0 milliliters (ml) double distilled HNO3 and 1.0 ml analytical grade H2O2. The solution was digested for 2 h then heated in a microwave oven to 180 °C for 10 min, then diluted with Milli-Q water. The concentrations of Al, As, Ba, Cd, Co, Cr, Cu, Fe, Mg, Mn, Mo, Ni, Pb, Se, Sr, V and Zn were measured with a Bruker 820-MS Inductively Coupled Plasma Mass Spectrometer (ICP-MS). An external calibration strategy was used for both instruments with a series of multi-element standard solutions containing all of the elements of interest (three standard solutions were tested containing 10, 25 and 50 µg L−1 of each element) and the results were reported after subtracting the procedure blanks. The multi-element standards were obtained from Choice Analytical (Sydney, Australia). Collisional Reaction Interface (CRI) was used for As (H2) and V (He), while 82Se isotope was used for Se quantification, to eliminate polyatomic interferences for these elements. A 1% HCl solution was spiked with 1 ppb As, Se and V and measured three times for quality control; recovery between 98.5 and 110% indicated no significant interferences. All analyses were conducted at the Advanced Analytical Centre at JCU, Townsville.

Statistical analysis

As the aim of the experiment was to compare long-term growth rates for the three species of Oedogonium in Ash Dam water with DTW, the mean growth rate was determined for each replicate over the course of the three week experimental period containing the ADW treatments. The mean growth rate was then analysed by two-factor analysis of variance (ANOVA) for the main experimental factors of water source (Ash Dam water and DTW) and species. Where necessary the growth data were log transformed (Quinn & Keough, 2002). Residual histograms and scatter plots of residuals vs. estimates were assessed to determine normality and homogeneity of variance respectively (Quinn & Keough, 2002). Post-hoc comparisons for main effects and their interactions were made using Tukey’s HSD multiple comparisons.

The concentrations of metals in the algal biomass were first subject to multivariate ordination with non-metric Multi-dimensional Scaling (nMDS) to assess multivariate patterns in the temporal profile of metal bioconcentration from Ash Dam water by the three species. The mean data for the metal concentration within the biomass (mg kg−1) were reassembled in a Bray-Curtis similarity matrix, focusing specifically on elements for which there are existing Australian and New Zealand Environmental Control Council (ANZECC) water quality criteria (see Table 1 for ANZECC water quality criteria). Mean metal contents of the three species were then contrasted for individual metals with a one-way ANOVA with the factor ‘species’. The elemental concentration of the algal biomass was also converted to a bioconcentration factor (BCF) for elements that have stipulated ANZECC trigger values. The BCF is the ratio of the element concentration in the water, to the final concentration in the biomass (DeForest, Brix & Adams, 2007). The bioconcentration factor (BCF) was calculated using the equation: BCF=Cb/Cw

where, Cb is the concentration of the element in the dry macroalgal biomass (mg kg−1) and Cw is the initial concentration of the element in the water phase (mg L−1). The BCF is therefore a dimensionless ratio that expresses uptake of elements relative to availability in the water sample.

Results

Elemental profile of Ash Dam water and DTW

The elemental composition of the Ash Dam water was complex with several metals and metalloids at concentrations significantly higher than the relevant ANZECC water quality criteria (Table 1). The concentrations of Al, As, Cd, Mg, Mn, Mo, Ni, Se, Sr, V and Zn were all significantly higher in Ash Dam water than DTW (two-tailed t test, P < 0.05), with the concentration of Al, As, Cd, Ni, Se and Zn exceeding the ANZECC trigger values designated for the protection of aquatic life at the 95% level (Table 1). The Ash Dam water is therefore a complex industrial effluent with multiple metal (Al, Cd, Ni and Zn) and metalloid (As and Se) targets for bioremediation. The remainder of the results section will focus on those elements that exceed ANZECC water quality criteria as the aim of the study is to ascertain interspecific variation in the potential for bioremediation.

Between-species variation in growth rates

‘Steady state’ productivity (defined here as <5% change in SGR for two consecutive weeks) was attained after 5 weeks of acclimation for all three species (Fig. S1). Acclimation to ‘steady state’ productivity was conducted to minimise any anomalous growth rates during the experiments as a result of changing photo-regime (µmol photons m−2 s−1) and photoperiod associated with the biomass being sourced from stock cultures and subsequently exposed to controlled experimental conditions. During this period, one of the Oedogonium species (KC701473) had a significantly higher mean productivity (23% SGR) than the other two species (∼20% SGR for both KF606977 and KC606974; Fig. S1).

After acclimation, all three species were exposed to either Ash Dam water or DTW with nutrient addition. There was a significant ‘species x water source’ interaction for the mean growth rates of the three species of Oedogonium (“Species x Water Source”: F2,12 = 4.95, P = 0.027). The growth rate of one species of Oedogonium (KC701473) was significantly higher than the growth rates of the two other species in DTW (Fig. 1). KC701473 had a mean SGR of 26% in DTW, in comparison to the other species that had a mean SGR of 20% (KF606977) and 16% (KC606974) over the three week experimental period (Fig. 1).

Figure 1 Specific growth rate of the three Oedogonium species in Ash Dam water and dechlorinated town water.

Data are means ± SE.

The overall growth rates for each species were reduced in Ash Dam water relative to those in DTW, with the greatest reduction for KC701473 (decreasing from 26% in DTW to 7.5% in Ash Dam water) (Fig. 1). However, in contrast to the significant difference observed in growth between the three species when grown in DTW, there was no difference in the growth rates between the three species of Oedogonium in Ash Dam water (Fig. 1). There was an apparent trend towards higher SGR for the Oedogonium from the Brandon region (KF606977) and Tarong (KC606974) than KC701473 in Ash Dam water (Fig. 1), but these differences were not statistically significant (Fig. 1).

Patterns in metal and metalloid bioconcentration

All three species had low concentrations of most elements at the end of the acclimation period in DTW, although KC701473 had a significantly higher K content than the other two Oedogonium species (Table S1). There were no significant differences between the three species of Oedogonium with respect to mean bioconcentration of metals or metalloids over the course of the experiment (Fig. 2). The only elements for which there was a significant difference in bioconcentration between species of Oedogonium were Cd and As. Cd was bioconcentrated to a significantly higher concentration in KF606977 and KC606974 than KC701743 (Fig. 3B, Table 2), while As was bioconcentrated to a significantly higher concentration in KC701743 than the other two species (Fig. 3E, Table 2). The remaining elements showed no significant differences in their bioconcentration between the three species of Oedogonium (Fig. 3, Table 2). The majority of elements that did not initially exceed ANZECC criteria also had similar concentrations in the three species of Oedogonium at the conclusion of the experiment (Fig. S2 and Table S2).

Figure 2 Mean concentration of (A) summed ANZECC metals and (B) summed ANZECC metalloids in the three Oedogonium species cultured in Ash Dam water at time 0 (black bars) and after three weeks of cultivation (grey bars).

Data are means ± SE.

Figure 3 Mean concentration of metals (Al, Cd, Ni and Zn) and metalloids (As and Se) in the three Oedogonium species in Ash Dam water at time 0 (black bars) and after three weeks of cultivation (grey bars).

Data are means ± SE.

Table 2 Analysis of variance for ANZECC metals (Al, Cd, Ni and Zn) and metalloids (As and Se) concentrations in the three Oedogonium species after cultivation in Ash Dam water.

Factor	df	Aluminium	Cadmium	Nickel	Zinc	Arsenic	Selenium	
		MS	F	P	MS	F	P	MS	F	P	MS	F	P	MS	F	P	MS	F	P	
Species	2	271.8	0.73	0.52	1.14	5.82	0.04	156.9	0.77	0.50	1141.6	3.66	0.09	178.1	7.41	0.02	10.1	0.67	0.55	
Residual	6	372.8			0.20			203.4			312.7			24.0			14.9			
Notes.

Bold values are statistically significant (P < 0.05).

While the mean bioconcentration of metals and metalloids did not differ between the three species by the end of the experiment, there were minor differences in the temporal pattern of metal bioconcentration. The clustering of the three species with respect to the vector orientation in the MDS supports a temporal sequence in metal uptake and that this differs between the three species. In the first week following the transfer of the cultures to Ash Dam water, all three species bioconcentrated a subset of elements from Ash Dam water (Mn, Pb and Fe, Fig. 4), however, KC606974 and KF606977 bioconcentrated the metals Cd, Ni and Zn much more rapidly in the first week than KC701473 (Fig. 4). By week two, KC701473 had a similar element profile to the other species having bioconcentrated the trace elements Al, Cd, Ni, Se and Zn, while KC606974 clustered with the group of element vectors of Se, Ni, Zn, Cd, Al and B and had the highest internal concentrations of these elements in week 2 (Fig. 4). In the third week of cultivation, the elemental profile of the Oedogonium KC606974 was similar to its initial (post-acclimation) composition, while the other two Oedogonium species (KC701473 and KC606974) continued to have higher concentrations of elements (Fig. 4).

Figure 4 Nonmetric multidimensional scaling bi-plot of biomass elemental profiles through time.

These patterns are also clearly reflected in the temporal sequence of bioconcentration of metals (Al, Cd, Ni and Zn) and metalloids (As, Se) that exceeded ANZECC criteria by the three species of Oedogonium. The four metals (Al, Cd, Ni and Zn) were bioconcentrated more rapidly by Oedogonium isolate KC606974 and KF606977 than KC701473. There were significantly higher internal concentrations of Al, Cd, Ni and Zn in these two species after one week of exposure to Ash Dam water (Figs. 5A–5D). However, the internal concentrations of Al, Cd and Ni in the three species converged and were not different by the end of the experiment, while Zn remained higher in KF606977 than the other two species (Fig. 5D). In contrast, bioconcentration of As differed between the three species, with significantly higher As contents in KC701473 than KF606977 and KC606974 by the end of the experiment (Figs. 5E) . There were no significant differences in the rate or extent of bioconcentration of Se by the three Oedogonium species (Fig. 5F).

Figure 5 Bioaccumulation of metals (Al, Cd, Ni and Zn) and metalloids (As and Se) by the three Oedogonium species from Ash Dam water through time.

Data are means ± SE.

Metal and metalloid bioconcentration factors

The element bioconcentration factors were very consistent between the three species of Oedogonium. For all species, the highest BCF was attained for Cu, Mn, Zn and Ni with values ranging from 709–1418 for Zn, 889–1256 for Ni, 6938–8943 for Cu, and 5892–16719 for Mn (Tables 3 and 4). Conversely, Al, Se and Mo all had the lowest BCF with values of 3–4 for Mo, 79–93 for Se and 259–351 for Al (Tables 3 and 4). The ranking of the BCF amongst elements was, however, relatively consistent with the same four elements (Cu, Mn, Zn and Ni) having the highest BCF for all three species, while the same three elements (Al, Se and Mo) had the lowest BCF for all species (Table 4).

Table 3 Bioconcentration factors for ANZECC-listed elements in Ash Dam water.

Element	Ash Dam water (mg L−1)	Bioconcentration factor	
		KF606977	KC701473	KC606974	
Al	0.1233	276	259	351	
As	0.0337	481	504	721	
Cd	0.0025	730	288	642	
Cu	0.001	6938	8943	8884	
Pb	0.001	346	438	1009	
Mn	0.004	5892	16719	12739	
Mo	1.28	3	3	3	
Ni	0.0347	889	929	1256	
Se	0.07	83	79	93	
Zn	0.055	1418	709	1033	

Table 4 Bioconcentration factors in order of decreasing magnitude.

Species	Bioconcentration factor	
KF606977	Cu > Mn > Zn > Ni > Cd > As > Pb > Al > Se > Mo	
KC701473	Mn > Cu > Ni > Zn > As > Pb > Cd > Al > Se > Mo	
KC606974	Mn > Cu > Ni > Zn > Pb > As > Cd > Al > Se > Mo	

Discussion

To be a suitable candidate for bioremediation applications, a target group of macroalgae should be widely distributed, robust to variable environmental conditions, be able to out-compete other species in intensive culture, and grow at high rates. The genus Oedogonium is already known to meet these criteria from individual studies investigating species within the genus (Lawton, de Nys & Paul, 2013). However, it remains unresolved as to whether individual species within the genus also have a similar capacity to grow and bioconcentrate metals and metalloids from waste water to provide a bioremediation service—given that there is substantial conspecific variation in growth rates and in metal concentrations for macroalgae more generally (Lawton et al., 2014). Our results clearly demonstrate that the individual species within the genus Oedogonium that were assessed in this study have an equal capacity to grow in a complex industrial waste stream, and to sequester metals and metalloids from this waste stream to provide a bioremediation service.

The specific growth rates (SGR) of Oedogonium averaged approximately 12% d−1 in complex waste water containing multiple elements that require remediation. Interestingly, all three species of Oedogonium had a similar SGR in Ash Dam water, despite quite significant differences in SGR when the same species were cultivated in DTW. These differences were evident during both the acclimation and experimental cultivation periods in DTW. The consistently lower growth rates of Oedogonium in Ash Dam water demonstrate that the growth rates of individual species in clean-water cultures cannot be used to predict their productivities in metal-contaminated waste waters, for which there was a relatively consistent reduction in SGR of approximately 50%. Furthermore, the species-level traits that confer more rapid growth rates in DTW do not confer more rapid growth rates in industrial effluent. The SGR of all species was higher in DTW, and for this reason site-specific evaluations of potential growth rates in industrial effluents will be required. In general, our data support the prediction that the species of the genus Oedogonium are strong candidates for metal-bioremediation programs given their ability to deliver consistent growth in complex effluents while accumulating metals and metalloids over time. However, this conclusion needs to be tempered in that individual species will always require trials to ensure their efficacy in complex waste streams, in ambient environmental conditions, and that the bioremediation of waste streams will, by nature of the composition of the waste stream and the endemic flora, be influenced by site-specific water quality profiles.

In addition to having similar growth rates, the three species of Oedogonium also had very similar patterns of metal bioconcentration. While there were minor differences in the temporal pattern of the bioconcentration of some elements between the three species, the mean bioconcentration of ANZECC-listed elements over the course of the experiments did not differ for the majority of elements. Each of the species was able to bioconcentrate internal metal and metalloid concentrations orders of magnitude above their concentrations in the Ash Dam water. The three species also had very similar elemental composition over the course of the experiment, and delivered consistent element-specific bioconcentration factors (BCFs) for each of the ANZECC-listed elements. The metals Cu, Mn, Zn and Ni were bioconcentrated to a significantly higher degree by the species of Oedogonium than the metalloids Mo and Se, supporting what has previously been published for single species (Saunders et al., 2012; Roberts, de Nys & Paul, 2013). This evidence adds to a clear consistency amongst studies that live algal cultures are effective at sequestering metal cations, but are less effective at sequestering metalloids that occur as oxyanions, particularly Se (as SeO42−), As (as AsO43−) and Mo (as MO42−) (Roberts, de Nys & Paul, 2013). The algal cell wall has a natural affinity for dissolved cations and this is exploited in the use of dried biomass as a biosorbent for dissolved metals (Kidgell et al., in press). In contrast, dissolved metalloids do not show a natural affinity for algal cell walls which may explain the limited uptake in these experiments. Additionally, there may be significant competition for uptake sites in the complex effluent tested in these experiments and the bioconcentration of the metalloids may be expected to increase in the absence of competing ions.

The only significant differences between species of Oedogonium with respect to the bioconcentration of elements were for the ANZECC metalloid As and the ANZECC metal Cd. KC701473 had a significantly higher content of As than the other two species, but a lower content of Cd. There is some evidence that the presence of As in contaminated soils can reduce the bioconcentration of Cd by terrestrial plants (Sun et al., 2009). Reduced Cd bioconcentration in the presence of As could be a result of competition for common binding sites on the algal cell wall by the two elements. Indeed, similar antagonistic relationships have been documented in marine seaweeds where the presence of Cu can impede Cd bioconcentration (Roberts, Johnston & Poore, 2008). The variance between the three species of Oedogonium with respect to As bioconcentration suggests it may be possible to identify species that are more effective at accumulating this metalloid, but there may be a trade-off of reduced Cd bioconcentration. There was also a slight difference in the temporal profile of metal bioconcentration, with the Tarong isolate KC606974 having rapid initial bioconcentration of the metals Al, Cd, Ni and Zn, followed by a slower rate of bioconcentration in the later stages of the experiment. This could be indicative of an efficient metal detoxification pathway in this isolate that was originally collected from Tarong Ash Dam. Green freshwater macroalgae can develop efficient metal detoxification mechanisms such as the production of metal-binding phytochelatins to sequester and detoxify contaminants in cellular vacuoles, and this property is most pronounced in isolates that have been exposed to contaminants for a long period of time (Pawlik-Skowrońska, 2001; Pawlik-Skowrońska, 2003). Regardless, all species had low BCFs for metalloids, and bioconcentrated them at a slow rate relative to their availability in the effluent, and this finding is consistent with previous research that has highlighted the relatively slow rates of metalloid bioconcentration by green macroalgae relative to the rates at which metals are bioconcentrated (Roberts, de Nys & Paul, 2013). Therefore, the three species of Oedogonium delivered equal growth rates and bioconcentrated the same elements preferentially such that the over-arching bioremediation potential of the three species was effectively the same. The genus Oedogonium therefore has a consistent ability to sequester a range of metals and metalloids from complex effluents for broad-spectrum remediation applications and, furthermore, it is possible to predict the key metals that will be most effectively remediated by conspecific species.

Importantly, we have demonstrated that species within the cosmopolitan algal genus Oedogonium have a consistent ability to bioconcentrate metals from waste water. One could therefore isolate native Oedogonium, potentially from the waste water source requiring remediation as we have done, and use these species to remediate waste water in intensive on-site cultivation. Notably, the Oedogonium KC606974 from Tarong used during this study is endemic at the Tarong power station (Stanwell Energy) and this provides an example of the capacity to use a native species for culture within the industrial waste stream (Ash Dam water from Tarong) to deliver the first step in a potentially larger integrated waste management process.

Supplemental information

Table S1 Concentration (mg kg−1) of elements in the three Oedogonium species before (initial) and after 1–3 weeks of cultivation in Ash Dam water. Data are means ± SE

Click here for additional data file.

Table S2 Analysis of Variance for individual elements in biomass that do no exceed ANZECC trigger values

Click here for additional data file.

Figure S1 Specific Growth Rate of the three Oedogonium species during the acclimation phase in dechlorinated town water. Data are means ± SE

Click here for additional data file.

Figure S2 Individual elements not in excess of ANZECC trigger values in the three Oedogonium species grown in Ash Dam Water at time 0 (black bars) and after three weeks of cultivation (grey bars). Data are means ± SE

Click here for additional data file.

Figure S2 continued Individual elements not in excess of ANZECC trigger values in the three Oedogonium species grown in Ash Dam Water at time 0 (black bars) and after three weeks of cultivation (grey bars). Data are means ± SE

Click here for additional data file.

We thank Stanwell Energy Corporation for providing effluents for the experimental studies and Yi Hu (Advanced Analytical Centre, James Cook University) for assistance with chemical analyses.

Additional Information and Declarations

Competing Interests

Author Contributions

This research is part of the MBD Energy Research and Development program for Biological Carbon Capture and Storage with the cooperation of Stanwell Energy Corporation. There are no further patents, products in development or marketed products to declare.

Michael B. Ellison conceived and designed the experiments, performed the experiments, analyzed the data, wrote the paper, prepared figures and/or tables.

Rocky de Nys conceived and designed the experiments, contributed reagents/materials/analysis tools, wrote the paper, reviewed drafts of the paper.

Nicholas A. Paul conceived and designed the experiments, analyzed the data, wrote the paper, reviewed drafts of the paper.

David A. Roberts conceived and designed the experiments, analyzed the data, wrote the paper, prepared figures and/or tables.

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
