# Peer review of "Growth and metal bioconcentration by conspecific freshwater macroalgae cultured in industrial waste water"

_PeerJ, doi:10.7717/peerj.401_

## Round 0.1 · original submission · Minor Revisions

The work is generally well-conceived and executed, and the presentation is clear but the reviewers have identified some issues. It should be possible to deal with the revisions without further experimental work, so my decision is "Minor Revision". All of the individual points raised by the reviewers should be addressed in detail in your revised manuscript and/or accompanying letter.

Of particular note:

(1) Some clarification of experimental design - e.g. source of organisms, choice of drying temperatures etc., is required.

(2) Include the statistical test and p value when a claim of significance is made in the text.

(3) Consider the recommendations from Reviewer 1 for expanding and clarifying your discussion, in particular ensure that statements are supported by the data and/or cited literature.

Reviewer 1 ·

Basic reporting

The content of the research article is interesting and well-researched. There are some parts of the manuscript that were unclear (please see general comments).
In general the manuscript is off to a good start, but needs some significant revisions prior to publication.

It is recommended that he authors add an explanation on the difference between metals and metalloids in the introduction.

Experimental design

Some clarification on methods and experimental design is recommended.

a) Line 71 -72. "....while the other two species were
collected from irrigation ditches in the Brandon sugarcane region."
Comment: When were the species from Brandon sugarcane collected?

b) Please elaborate on the experimental design. Line 105 reads "for three consecutive growth periods of seven days".
Comment: This is a little unclear. Do you mean each growth period was 7 days and you stopped and started the experiment 3 times? So the total number of days of incubation was 21 days?

c) Line 113-115 "Each replicate bottle was harvested every seven days with a complete water exchange and all replicates (n = 3) were replenished with new f/2 growth media. All replicates were harvested and the biomass blot-dried with paper towel and weighed to the nearest 0.1 g."
Comment: This section is a bit repetitive and somewhat unclear. Maybe clarify by saying "The algae in each replicate bottle was harvested every seven days. The water from each Schott bottle was removed (how? just poured out? or did you use a pipette?) and then where did it go (did you send it to a lab for analysis?)"

d) Line 118: "All the dried biomass was stored at 4.0°C for further analysis."
Comment: Did you combine all of the dry algae at the end of the 3rd growth period? Or did you analyze the concentration of metal in each one separately? What was the "further analysis"? Was there any acid digestion performed on the algae? Did the lab that analyzed your algae use LCMS? HPLC? ICP-MS?

d) Line 131 "and at each of the three time points during Ash Dam water cultivation".
Comment: Do you mean during ADW "replacement" instead of "cultivation"?

Validity of the findings

Clarification is recommended in regards to statistical significance. If a statement suggests a significant difference, it is recommended that the authors put the statistical test used and the p value of the test in parenthesis at the end of each statement Example: (ANOVA, p<0.001). Even, if the statistics are not significant, one should still right the test and p value in parenthesis following the statement.

a)Examples:
Lines 168-169: "The concentrations of Al, As, Cd, Mg, Mo, Ni, Se, Sr, V and Zn were all significantly higher in Ash Dam water than DTW..."
Comment: Based on what statistic? ANOVA? Was p<0.05?

Lines 185-186. "There was a significant ‘species x water source’ interaction for the mean growth rates of the three species of Oedogonium (Table 2)."
Comment: Statistical test and p value should be included in parenthesis after the sentence (ANOVA, p<0.001).

Lines 193-195: "However, in contrast to the significant difference in growth in DTW, there was no difference in the growth rates of the three species of Oedogonium in Ash Dam water (Figure 1)."
Comment: Unclear. Maybe restate as "in contrast to the significant difference observed in growth between the three species when grown in DTW, there was no difference in the growth rates between the three species of Oedogonium in Ash Dam water" (Statistical test, p value).

Lines 203-204: "The only elements for which there was a significant
difference between species of Oedogonium were Cd and As"
Comment: The authors forgot to mention BCF in statement. Maybe it should read: "the only elements for which there was a significant difference in BCFs were Cd and As" (Statistical test, p value) (Figure 3).

Line 207-208: "The remaining elements showed no significant
differences in their bioconcentration by the three species of Oedogonium. (Figure 3, Table 3)."
Comment: Unclear statement. Reword. Maybe change the word "by" to "between"?

b) Lines 209-211. "The same pattern was also seen for the majority of elements that did not initially exceed ANZECC criteria, most of which had similar concentrations in the three species of Oedogonium at the conclusion of the experiment (Supplementary Materials Figure S1 and Table S2)."
Comment: Statement unclear. Please elaborate on what is meant by the "pattern".

c)Line 214. What is meant by a "vector" in the MDS plot? The authors may want to mention briefly how a Bray-Curtis plot is interpreted.

d) Line 240-241. "the highest BCF was attained for Cu, Mn, Zn and Ni with values
ranging from 709-1418 for Zn, up to 5892-16719 for Mn"
Comment: What about Cu and Ni?

e)Lines 241-242 "Conversely, Al, Se and Mo all had the lowest BCF with values of 3-4 for Mo up to 259-351 for Al"
Comment: What about for Selenium?

f)Line243. "The ranking of the BCF amongst elements was, however, relatively consistent."
Comment: The authors should discuss Table 5 a bit more.

4)Comments on Figures and Tables
In general, it is recommended the authors add more information to their figure and table legends. It is suggested that they add the sample size to each legend, what statistical tests (if any) were used, and what the various letters in each figure represent. Capitalization is not consistent in the legends.

a)In Figure 1, it is recommended that the authors add the abbreviations ADW and DTW for Ash Dam Water and Dechlorinated Town Water to the figure legend for clarity. The sample size and length of algae incubation should be included in the legend.

b)In the legends of Figures 2 and 3 you may want to clarify what "t (0)" refers to.

c)What do the letters "a" and "b" mean in Figure 3? Add to legend.

d) Each graph in Figure 5 should be labeled "a", "b", "c", and "d" as they are referred to as such in the text (lines 231-237).

e) Table 2 title "Analysis of variance for the Specific Growth Rate ...." should be modified. Variance, specific, growth, and rate should not be capitalized.

Discussion
a) Line 254. The author uses the terms "versatile" and "robust". Can the authors elaborate what is meant by these terms?
b) Line 263. What is meant by "appropriate proxy"?
c) Line 270 - 274. A bit confusing. What is meant by "bespoke".
d) Line 287 . The authors should discuss and reference why they think the alga concentrates metalloids to a lesser extent than metals? Is it because maybe the negatively charged cell wall of the alga repels metalloids?
e) Lines 291-294. "KC701473 had a significantly higher content of As than the other two species, but a lower content of Cd. Previous research has highlighted the relatively slow rates of metalloid bioconcentration by green macroalgae relative to the rates at which metals are bioconcentrated (Roberts et al. 2013)."
Comments:1) Why do you think KC701473 had more As and less Cd? Is it possible that the arsenic blocked the ion channels by which cadmium is taken up? Do you think the fact that this alga had a high concentration of K affected Cd uptake?
2)The citation by Roberts following the statement comparing As and Cd uptake isn't supporting the previous statement. Although, it may be true that some macroalga concentrate metals at slow rates, how does this relate to the Cd and As data? It is recommended that the authors find a more relevant paper explaining metal transport in algae.
f)Lines 297-300. "Therefore, the three species of Oedogonium delivered equal growth rates, equal rates of metal bioconcentration, and bioconcentrated the same elements preferentially such that the overarching bioremediation potential of the three species was effectively the same."
Comment: This statement is not true. The author's did not show data on the rate of bioconcentration (metal uptake over time). Their data suggest differences in bioconcentration factors for As and Cd between the species of Oedogonium, so the alga did not bioconcentrate the elements identically.
g) In Figure 5, the authors may want to discuss why the species KC606974 shows increased bioconcentration of Al,Cd,Ni, and Zn during week 2, but a decreased bioconcentration at week 3. Does the fact that this species was adapted to Tarong's environment have anything to do with its response?
It would be interesting to see if there are any molecular changes over time in each of the species. Maybe genes involved in metal tolerance are expressed over time? If so, maybe this would make the alga even more beneficial in bioremediation?

Additional comments

The question on whether the three collected species of Oedogonium are suitable for bioremediation of industrial waste water is an important one. The authors had a valid experimental design and obtained bioconcentration factors of metals and metalloids commonly found in the wastewater of interest. Statements expressing statistical importance should have the appropriate statistical citation after the statement (Statistical test, p value). It is also suggested that the authors elaborate on their discussion of metal transport in macroalgae and incorporate the suggested modifications into their manuscript.

·

Basic reporting

The work conducted is very valuable and useful in the bioremediation of polluted industrial waste water. Valuable insight of the growth rates as well as the bioconcentration of metals/metalloids on Oedogonium species in different media (ash dam water and dechlorinated town water) are also provided.

Experimental design

Experiments conducted are clearly described.
The cultivation study of the algae was conducted at 28oC,is there any particular reason for choosing this temperature?( line 98)
Which analysis was performed with the biomass dried at 60oC, (line118), whereas elemental analysis was performed in the biomass dried at 48oC (line 132)?
What is the specification of the multi-element standard used for metal analysis?

Validity of the findings

The conclusion of the findings are very relevant.

Additional comments

In Figure 1, a and b are not defined.
The difference in SGR between KF606977 and KC701473 in ADW seems to be significant, as it is greater than 5% as shown in Figure 1.

---

## Round 0.2 · accepted · Accept

Thank you for your comprehensive response to the reviewers' comments.